# Physical and Numerical Simulation of the Mechanism Underpinning Accumulation Layer Deformation, Instability, and Movement Caused by Changing Reservoir Water Levels

Rubin Wang [1],*, Jianxin Wan [2], Ruilin Cheng [3], Yunzi Wang [2],* and Zhaoying Wang [4]

1   Research Institute of Geotechnical Engineering, Hohai University, Nanjing 210098, China
2   College of Civil and Transportation Engineering, Hohai University, Nanjing 210098, China;
    wjx13141206@163.com
3   Guiyang Engineering Corporation Limited, PowerChina, Guiyang 550081, China;
    chengrl_gyy@powerchina.cn
4   Huaneng Lancang River Hydropower Inc., Kunming 650206, China; rmwangzhaoying@sohu.com
*   Correspondence: rbwang_hhu@foxmail.com (R.W.); wyz19980516@163.com (Y.W.)

**Abstract:** Large-scale physical models of landslides can potentially accurately reflect the interactions between many internal and external factors and elucidate the process of slope deformation and failure. In order to reveal the mechanism of deformation of the reservoir bank accumulation layer, in this study, a large-scale physical test model with a similarity ratio of 1:200 was constructed based on the actual engineering geological section. Two reservoir water level cycle fluctuation conditions were simulated, and the reservoir water level drop rate was strictly controlled to be two times the rise rate. This study analyzed pore water pressure and deformation characteristics in the accumulation layer in relation to fluctuating reservoir water levels. The results showed that the rise in reservoir water level will make landslides more stable. The periodic sudden drop in water level seriously endangers the stability of landslides. The deformation and failure of landslides are more likely to occur in the weak interlayer area. The failure mode of the accumulation body in the test was traction failure. It is suggested that the front part of the accumulation body can be reinforced in practical engineering. To reveal the progression of instability and movement during accumulation layer large-scale landslides, a numerical model was constructed using the material point method. The accumulation layer sliding process could be divided into three stages: acceleration sliding, deceleration sliding and stabilization. After destabilization, the river channel may be altered by the landslide mass to form a landslide dam, potentially threatening the integrity of the dam via impulse waves generated during destabilization. The research results provide technical support for reservoir scheduling in major water conservancy and hydroelectric power station reservoirs as well as engineering risk assessment and prevention.

**Keywords:** reservoir water level fluctuation; physical model test; deformation mechanism; material point method; accumulation layer landslide

---

## 1. Introduction

Reservoir impoundment greatly influences the stability of a bank slope, and changes in the reservoir water level often induce slope instability and damage, resulting in landslides. For example, the Vajont landslide in Italy, Tangyanguang landslide in the Tuoxi Reservoir in Hunan Province, Qianjiangping landslide in the Three Gorges Reservoir Region, and Hongyanzi landslide occurred during impoundment.

Water-induced landslides are generally associated with quaternary deposits. When the water level changes, penetration of the sliding body is poor, and the seepage or discharge rate of the accumulation layer is slow. The effect of reservoir water on such landslides is mainly linked to changing water pressure. The changing water pressure generated by reservoir water level fluctuation in the sliding body affects the overall stability of the accumulation layer [1].

Moreover, worldwide dam construction over the past century has shown that the water storage period of large reservoirs leads to bank slope instability and deformation. Therefore, in-depth research on variable deformation and destruction mechanisms under different reservoir water fluctuations is needed for the construction and safe operation of hydroelectric power stations. Numerous studies have analyzed these mechanisms using methods such as on-site monitoring, numerical simulation, and physical models. The potential effects of rainfall intensity and reservoir water level on reservoir landslide displacement are mainly studied by analyzing monitoring point displacement [2–4], and the numerical simulation [5–7] and physical model tests [8–11] are the important means for simulating the slope stability caused by rainfall and reservoir impoundment.

At present, on-site monitoring, early warning technology, and numerical simulation of landslide movement cannot fully elucidate the mechanism of reservoir bank instability and associated landslides induced by hydrodynamic pressure. However, large-scale physical models of landslides can consider and accurately reflect the interactions between many internal and external factors and intuitively illustrate the entire process of landslide deformation and failure. Therefore, this is an effective approach to analyzing the evolution of landslides and the mechanism underpinning slope instability. In terms of physical experiments with the accumulation body, most scholars consider the stability of the accumulation body caused by rainfall. Few scholars have studied the stability of the accumulation body under the sudden drop of the reservoir water level, and fewer scholars have studied the stability of the accumulation body under periodic change of the reservoir water level [12,13]. In this study, two cycles of reservoir water level change were designed, and the speed of water storage was controlled to be half the speed of water level decline. At present, this test scheme is still rare. In the aspect of numerical simulation, most study the deformation of landslide caused by rainfall [14], and the study of landslide instability caused by periodic water level change is also rare. The material point method (MPM), used to simulate the evolution of landslide instability and movement in the accumulation layer of the reservoir bank, can obtain not only the initial mode of landslide instability but also the path and accumulation form of the landslide, which is helpful in assessing the risk of dam bursting after a reservoir bank landslide [15–18]. There are also some shortcomings in the model test and MPM method. The preparatory work of the model test is large, the cost is high, and the personnel labor is heavy. The preparation process of the material is complex under the condition of ensuring the similarity ratio of the model. It is possible to draw conclusions that do not match reality, so the cost of the model test is high. The smooth development of the test may be limited. As for numerical simulations of the pinning accumulation layer, the existence of the sliding surface is preset, and only the flow process after landslide instability is focused on. The calculation cost is relatively large, and the external load disturbance of rainstorm erosion cannot be considered in the calculation. Without considering the start-up and failure mechanism of the actual landslide, as well as the influence of cracks, joints, weak interlayers, etc., it is impossible to determine the location of the sliding instability surface and the failure mechanism. In this study, a physical model test was conducted by selecting the geological profile of a typical accumulation layer to simulate the fluctuation in reservoir water level. This test was used to obtain data on the entire process of landslide deformation and evolution, displacement, pore water pressure, and other variables. By analyzing the deformation characteristics, instability mode, and failure mechanism of the accumulation layer, establishing an MPM numerical model based on the physical model test results, and predicting the evolution of accumulation layer instability using the MPM, this study aimed to serve as a reference point for future landslide research as well as engineering risk assessment and prevention.

## 2. Materials and Methods

### 2.1. Description of Accumulation Layer Area

The front edge of the accumulation layer modeled in this study is located deep in the Lancang River in China, and the back edge and sides comprise bedrock. The location of

this accumulation layer is shown in Figure 1. The elevation of the layer is 2654–3400 m, the slope angle is 32–37°, and the volume is $4.7 \times 10^7$ m$^3$. When the reservoir water level reaches the normal storage level of 2895 m, 243 m of the layer falls below the water level. Geological surveys have shown that the layer was mainly formed by the collapse of soil and rock, with a small amount of sand and gravel present in the lower part, and the bedrock is composed of Middle Triassic dacite. Drilling exploration results have shown that the layer is 10–100 m thick, and a typical geological section is shown in Figure 2.

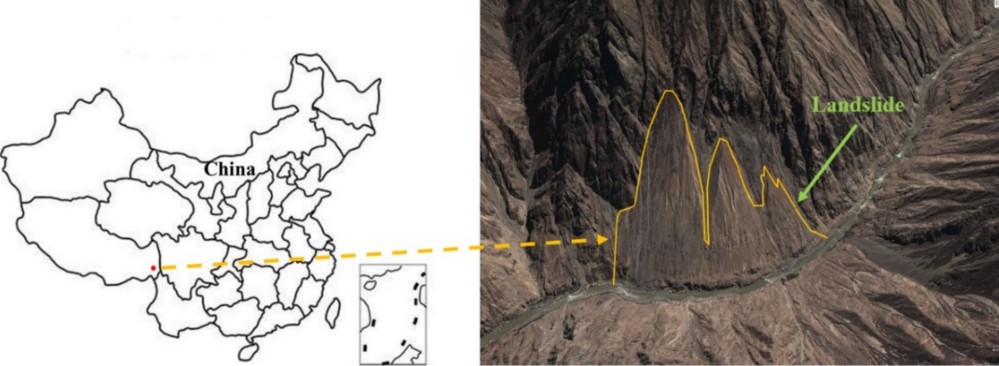

**Figure 1.** Location of the modeled accumulation layer.

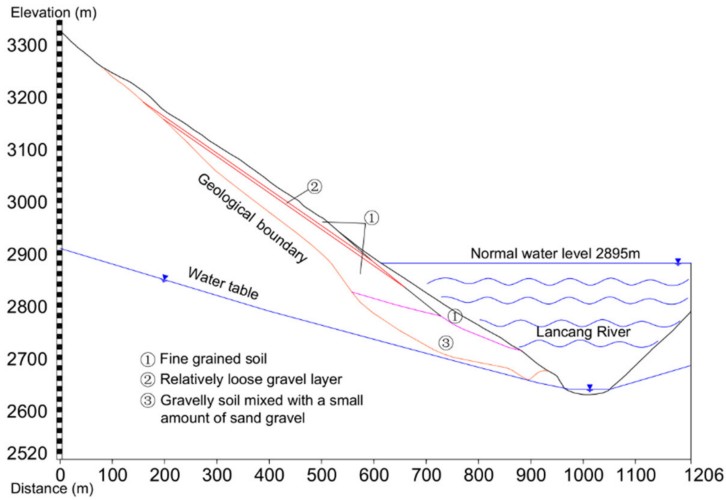

**Figure 2.** Typical profile of the Lancang River accumulation layer.

Before impoundment, the groundwater level of the accumulation layer was higher than that of the Lancang River. The groundwater in the layer is mainly replenished by atmospheric precipitation, and surface discharge conditions are good. Bedrock fissure water is mainly found in the dacite fissures and is fed via upper pore infiltration and precipitation. The maximum rainfall in the area is 55 mm. Overall rainfall is relatively low, with annual average rainfall exceeded by evaporation.

### 2.2. Physical Model of the Accumulation Layer Landslide

### 2.2.1. Model Test System

The external frame of the landslide model was 8 m long, 1 m wide, and 4.5 m high. As shown in the design diagram in Figure 3, the model was mainly composed of an external frame (1), reservoir water level control system (2), data monitoring and acquisition system, consisting of four displacement sensors (D1–D4) and three pore water pressure sensors (P1–P3) (3), and camera (4). The test model mainly consisted of bedrock and a sliding body. In the study, due to the test conditions and the geometric size of the prototype landslide,

the geometric similarity ratio was determined to be 1:200. The bedrock in the test was simplified as impermeable, so a cement mortar surface was used to simulate the bedrock surface. The configured soil was layered into the corresponding accumulation material area and compacted.

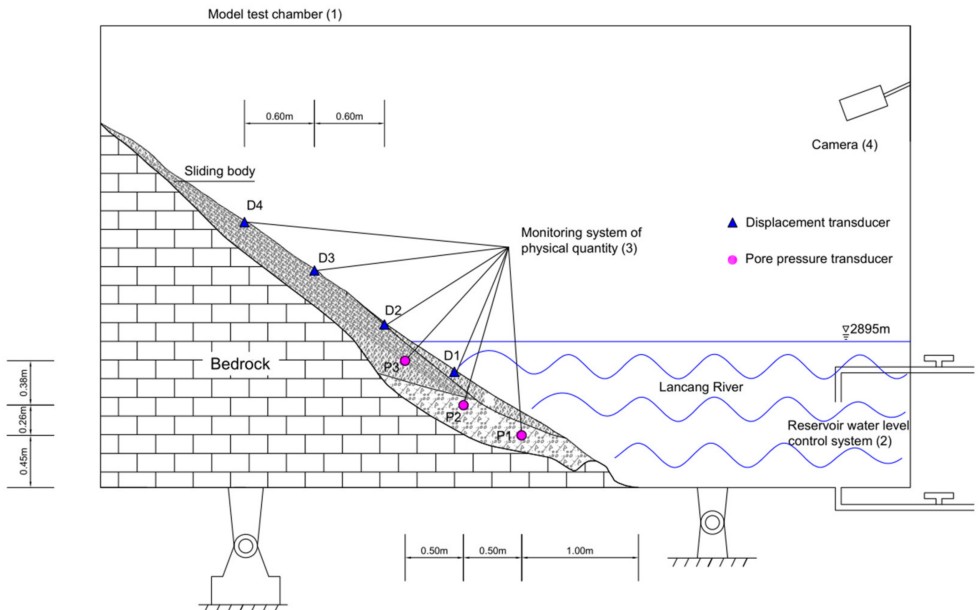

**Figure 3.** Schematic diagram of the landslide test model.

### 2.2.2. Similar Materials

Dimensional analysis and similarity theory were used to determine the similarity of the landslide model at different reservoir water levels. The main parameters of similar materials in the physical landslide model included the geometric similarity ratio, density, cohesion, internal friction angle, and permeability coefficient. According to similarity theory, similar materials in accumulation layers are isotropic. The gravity acceleration similarity ratio ($C_g$), density similarity ratio ($C_p$), and bulk density similarity ratio ($C_\gamma$) of this physical landslide model were 1, and the geometric similarity ratio ($C_l$) was 200. Based on dimensional analysis, according to the "homogeneous theorem", the parameters related to the problem are expressed in the same equation.

$$f(l, \rho, g, c, \varphi, E, u, k, \sigma, t, \mu, \theta, q, p) = 0 \tag{1}$$

$\varphi$ (internal friction angle), $\mu$ (Poisson's ratio), $\varepsilon$ (strain) are dimensionless quantities, $l$ (length), $u$ (displacement) are the same dimensions, $c$ (cohesion), $E$ (elastic modulus), $\sigma$ (stress) are the same dimensions, and $k$ (permeability coefficient), $v$ (flow velocity), $t$ (time) are the same dimensions. According to the dimensionless similarity ratio of 1, the same dimension has the same similarity ratio. So, we can obtain the similarity ratio of each physical quantity. The similarity ratio ($C$) of displacement ($u$), deformation modulus ($E$), cohesion ($c$), time ($t$), Poisson's ratio ($\mu$), internal friction angle ($\varphi$), permeability coefficient ($k$), and reservoir water level fluctuation speed ($v$) are related as follows: $C_u = C_E = C_c = 200$; $C_t = C_k = C_v = \sqrt{200}$; and $C_\mu = C_\varphi = 1$. The weak surface of the accumulation layer was composed of sand and soil, and the slippery material was composed of crushed stone, sand, soil, and pebbles.

Table 1 lists the physical and mechanical parameters of similar and original materials obtained by the homogenization design method model test, and Figure 4 shows the particle gradation curves of the prototype and model materials of the accumulation layer.

**Table 1.** Physical and mechanical parameters of the experimental model.

|  | $c$ (kPa) | $\varphi$ (°) | $\rho$ (g/cm$^3$) | $k$ (m/s) | $\gamma$ (kN/m$^3$) |
|---|---|---|---|---|---|
| Slide prototype material 1 | 50.0 | 33.0 | 2.10 | $8.57 \times 10^{-6}$ | 21.5 |
| Test model material 1 | 0.25 | 34.2 | 2.10 | $5.98 \times 10^{-7}$ | 21.5 |
| Slide prototype material 2 | 20.0 | 35.5 | 2.15 | $1.93 \times 10^{-4}$ | 21.5 |
| Test model material 2 | 0.1 | 36.4 | 2.15 | $1.38 \times 10^{-5}$ | 21.5 |

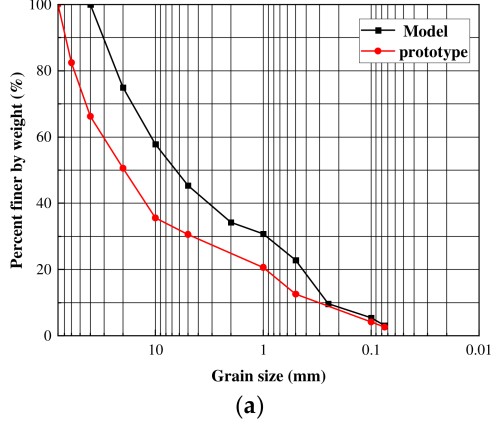
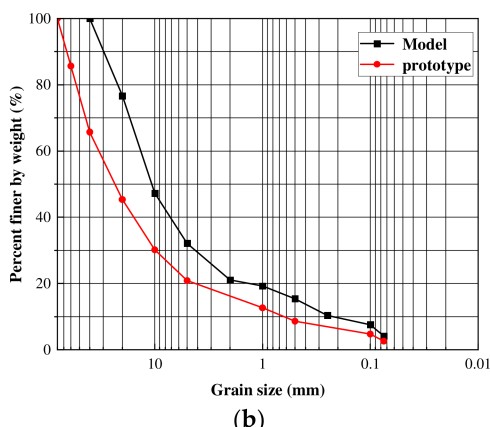

(a)          (b)

**Figure 4.** Particle size curves of prototype of accumulation layers and model materials. (**a**) Material 1; (**b**) Material 2.

### 2.2.3. Instrument Layout

A typical geological section of the accumulation layer was selected for the physical model test. The three pore water pressure sensors and four displacement sensors were installed at different points in the layer (Figure 3). The area affected by fluctuations in the reservoir water level was mainly located at the front edge of the accumulation layer; therefore, the pore water pressure sensors were mainly arranged at the front edge.

### 2.2.4. Test Procedure

Based on the rising rate of the water level in the reservoir area of the hydroelectric power station, the water level rising rate of the model test was set to 5.9 cm/h. To determine the effect of the rapid decline in water level caused by reservoir flood discharge, continuous rainfall, and other extreme conditions on hydrodynamic pressure landslides, the rate of water level decrease in the experimental reservoir was set to 11.8 cm/h. We designed two water level cycle test conditions according to the actual situation; in the second cycle, the reservoir water storage speed and water level drop speed were the same as those in the first cycle, and the reservoir water drop speed in both cycles was 2 times the water level rise speed during storage. The specific steps of the test procedure are detailed in Table 2. Based on these changes in water level, through the pre-buried pore hydraulic sensor connected to the computer, the change value of pore water pressure could be obtained in the acquisition system of the computer. When the landslide displacement occurred, the cable displacement sensor transmitted the displacement change to the acquisition software of the computer, and the change in the displacement value could be obtained. Pore water pressure and displacement were measured at each of their respective measuring points to obtain associated change curves.

**Table 2.** Experimental changes in water level over time.

|  | Speed (cm/h) | Duration (h) | Maintained Time (h) |
| --- | --- | --- | --- |
| Rise | 5.9 | 20.6 | 6.8 |
| Fall | 11.8 | 10.3 | 6.8 |
| Rise | 5.9 | 20.6 | 6.8 |
| Fall | 11.8 | 10.3 | 6.8 |

### *2.3. Numerical Model of the Accumulation Layer Landslide*

Based on the on-site geological survey and physical model test results, the topographical data and sliding surface of the accumulation layer were obtained. These were then used to construct an MPM numerical model to analyze the evolution of the sliding process of the accumulation layer.

Numerical Model Description

Based on the on-site geological survey and physical model test results, the MPM numerical model was constructed. The evolution of velocity and displacement fields during the movement of the accumulation body was studied and analyzed, and the deposition calculation results of the accumulation body were compared with the model test. The model was divided into three parts: the landslide body, sliding bed, and sliding surface. The background calculation grid was $3 \times 3$ m in size and composed of triangular elements, each of which contained one material point. The model had a total of 22,714 elements, 11,659 nodes, and 3345 material points. Each material point included the mass, volume, and related parameters of its associated region. The Mohr–Coulomb model was used to simulate the accumulation layer and sliding surface, whereas the elastic model was used to simulate the sliding bed. The calculation parameters of the MPM model are listed in Table 3, and the resulting model is shown in Figure 5. In order to study the instability, failure, and sliding path of the accumulation layer, six monitoring points were set up inside the layer to measure sliding velocity and displacement fields. The arrangement of the six velocity and displacement monitoring points is shown in Figure 6.

**Table 3.** Parameters for MPM model calculations.

| Mechanical Parameters | $c$ (kPa) | $\varphi$ (°) | $E$ (MPa) | $\mu$ | $\gamma$ (kN/m$^3$) |
| --- | --- | --- | --- | --- | --- |
| Landslide | 50.0 | 33.0 | 100 | 0.34 | 21.5 |
| Sliding bed | — | — | 10,000 | 0.34 | 21.5 |
| Sliding surface | 0 | 23.1 | 100 | 0.34 | 21.5 |

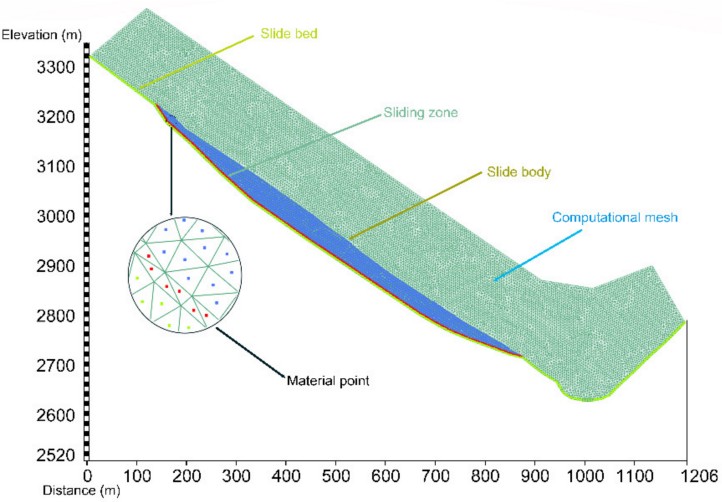

**Figure 5.** MPM model of the accumulation layer.

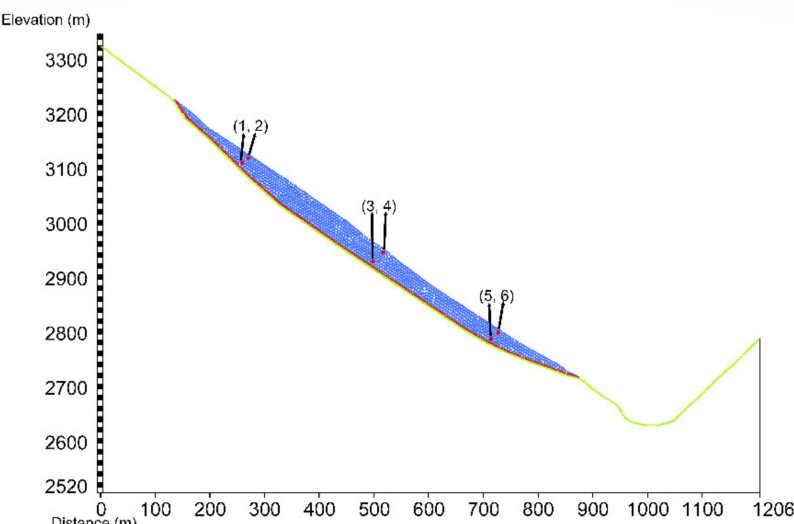

**Figure 6.** Layout of velocity and displacement monitoring points in the MPM model of the accumulation layer.

The initial equilibrium local damping was set to 0.70 before and 0.03 during landslide movement, respectively. Boundary conditions were simulated by constraining the normal displacement of the left, right, front, and rear boundaries of the model. The displacement of the bottom boundary was constrained to prevent vertical and horizontal displacement of the model. Finally, vertical constraints were imposed on the top of the mesh.

## 3. Results and Discussion

### 3.1. Test Results and Analysis

3.1.1. Variation in Pore Water Pressure of the Accumulation Layer

As shown in Figure 7, according to the change in pore water pressure, the whole test process was divided into six parts. I represents the pore water pressure in the first period of the test, which increased to a stable stage. II represents the stable stage of pore water pressure in the first cycle of the test. III represents the pore water pressure drop stage in the first cycle of the test. IV represents the pore water pressure in the second cycle of the test, increasing to a stable stage. V represents the stable stage of pore water pressure in the second cycle of the test. VI represents the pore water pressure drop stage in the second cycle of the test. As the reservoir water level was increased to its normal level, the pore water pressure value measured by sensor P1 at the front edge of the accumulation layer changed markedly and increased rapidly. As the water level continued to rise, reservoir water continued to infiltrate the slope. The pore water pressure and water level inside the accumulation layer also increased. The lower soil was gradually saturated, whereas the upper soil remained unsaturated. As such, the pore water pressure in the lower region of the accumulation layer was greater than that in the upper region.

While maintaining the normal reservoir water level, the maximum pore water pressure fluctuated slightly because of uneven deformation caused by water immersion and soil softening. When the water level subsequently decreased, the rate at which water pressure decreased lagged behind that of the reservoir water level. When the water level decreased to the point where the natural water level remained unchanged, pore water pressure in the accumulation layer continued to slowly decrease. The smaller the decrease in pore water pressure in the accumulation layer, the smaller the falling range of pore water pressure, and the greater the difference between pore water pressure inside and outside the accumulation layer [19].

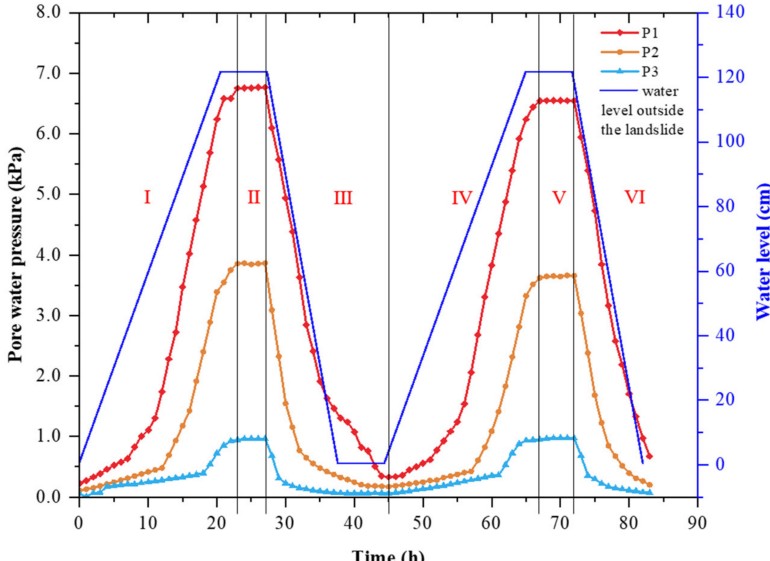

**Figure 7.** Pore water pressure changes at different sensors (P1–P3) in relation to reservoir water level fluctuations.

In the process of water level rising, when the water level rose to the highest point, the pore water pressure at each monitoring point continued to increase until it was stable. The reason is that the permeability of the accumulation soil hindered the infiltration of the water body, so the maximum value of pore water pressure appeared later than the time when the water level reached its highest point. When the reservoir water level dropped to the lowest height, the pore water pressure continued to decrease slowly. The pore water pressure in the slope showed a significant delay effect relative to the decline in the water level outside the slope. This delay effect caused water to drive soil particles to flow outward to the slope surface, which had a serious impact on the stability of the slope [20].

After the first test involving reservoir water level change, the water content in the accumulation layer was relatively high; therefore, the pore water pressure sensors P1 and P2 responded rapidly to the increase in water level. When the water level was increased to its normal level for the second time, the peak values recorded at P1 and P2 were slightly lower than those corresponding to the first peak. The main reason for this was the presence of deformation and tension cracks in the accumulation layer after the first fluctuation in the reservoir water level.

### 3.1.2. Cumulative Displacement of the Accumulation Layer

As shown in Figure 8, the cumulative displacement of the accumulation layer model changed with reservoir water level fluctuations. The closer the monitoring point was to the bottom of the landslide body, the greater the landslide displacement. When the reservoir water level increased and remained at its normal level, the displacement of the accumulation layer exhibited a slow creep. When the reservoir water level subsequently decreased sharply at a rate of 11.8 cm/h, the cumulative displacement and displacement rates at measurement points D1–D4 increased markedly. Based on recorded sensor values, both cumulative displacement and the displacement rate were highest as D1, followed by D2, D3, and D4. Overall, the results of the two test phases showed that when the reservoir water level increased rapidly, accumulation layer displacement increased slowly, whereas when that level dropped sharply, displacement increased rapidly [21].

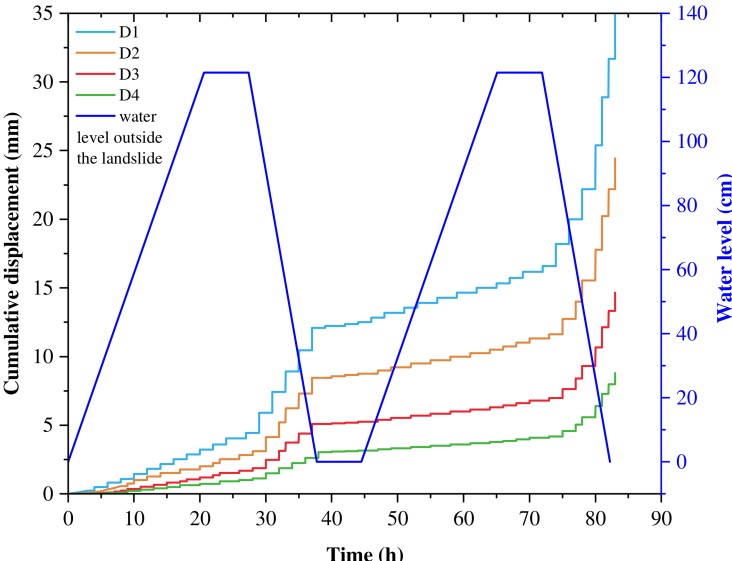

**Figure 8.** Cumulative displacement changes under reservoir water level fluctuations.

The displacement at each monitoring point after the first water level increase was compared with that of the second water level increase. The water body penetrated into the accumulation layer. Due to the softening and argillization of the water body, the shear strength of the soil was reduced, and the soil was more prone to deformation and failure. Therefore, the displacement at the monitoring point during the second water storage process was large, and the accumulation of displaced soil produced a potential slip surface and a decrease in the shear strength of the soil. When the water level dropped the second time, the shear strength of the soil was not enough to support the sliding force of the soil, resulting in an obvious overall slip of the accumulation layer along the weak interlayer [22].

### 3.1.3. Deformation Mechanism and Failure Characteristics of the Accumulation Layer

The failure characteristics of the accumulation layer in relation to changing reservoir water levels during the test are shown in Figure 9. The main failure characteristics of the landslide were three tensile cracks. During the test, partial disintegration of the accumulation layer first occurred at the front edge and then progressed along the cracks. Displacement was closely related to the deformation of the layer. The local sliding phenomenon at the front edge of the layer enlarged the empty surface, causing the front edge of the layer to lose support. However, the seepage path was shortened, the seepage process was strengthened, and the change in pore water pressure intensified. Because of the sudden drop in the reservoir water level, the fractures at the front edge of the accumulation layer extended and connected with one another, and local sliding gradually intensified. Thus, the two processes promoted each other. Tension cracks also began to appear in the middle and trailing edges of the layer, and weak interlayer cracks developed and expanded until they penetrated the layer. Finally, the front edge of the layer slipped first.

During the test, pore water pressure in the accumulation layer increased markedly as the reservoir water level rose, and decreased rapidly as the reservoir water level dropped. During the sudden drop in the water level, the infiltration line inside the layer manifested as a steep drop and bend lower down on the outside of the accumulation layer, whereas the inside of the layer exhibited a gradual slope. The further inside the layer, the closer to the water level infiltration line before the decline, as shown in Figure 10. The faster the reservoir water level dropped, the smaller the permeability coefficient of the accumulation layer's rock mass, the slower the groundwater level dropped, the steeper the groundwater level infiltration line near the outer edge of the accumulative layer, and the greater the gap between the reservoir water and groundwater levels.

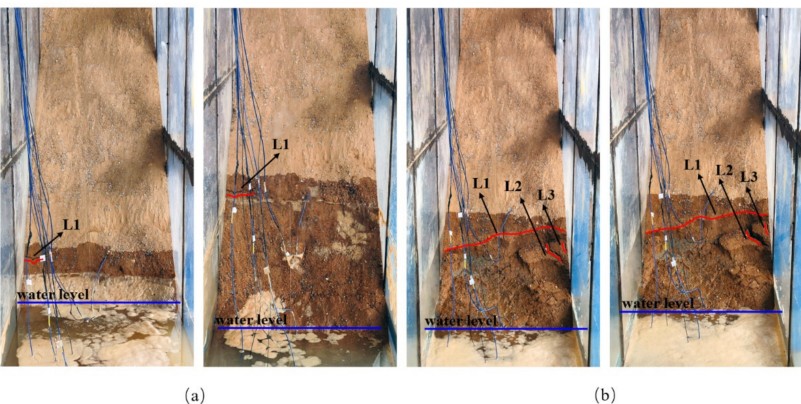

(a)            (b)

**Figure 9.** Failure characteristics of the accumulation layers during (**a**) the first and (**b**) the second phase of the test.

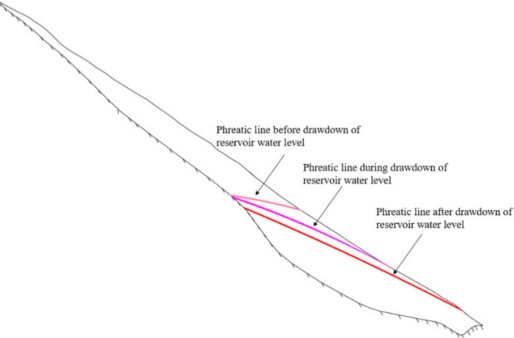

**Figure 10.** Schematic diagram of groundwater seepage in the accumulation layer of the reservoir.

The decrease in the groundwater level of the accumulation layer lagged behind the rapid decrease in the reservoir water level, resulting in a positive difference between the latter and the former. Because of osmotic pressure, groundwater seeped out towards the exterior of the accumulation layer. During the sudden drop in the reservoir water level, in addition to this osmotic pressure, high dynamic water pressure acted on the accumulation layer. After the reservoir water level decreased and stabilized for the first time, seepage in the layer remained relatively stable. The seepage pressure caused by the change in the reservoir water level gradually decreased to zero, and the stability of the accumulation layer increased slightly during this period. During the second phase of rising and falling reservoir water level, the continuous action of the water and the associated softening of the accumulation layer soil resulted in a decrease in the internal friction angle. Meanwhile, the cohesion of cemented material in the soil body decreased, such that the shear strength of the soil at the front edge of the accumulation layer decreased because of the decreasing reservoir water level. The front edge of the layer was thus damaged, resulting in a deformation and failure mode known as "traction-type" sliding.

### 3.2. Simulating the Evolution of Instability and Landslide Movement in the Accumulation Layer Analysis of Results

The simulations of the sliding process of the accumulation layer are shown in Figure 11. Different from previous studies, they were classified according to the change in acceleration. We classified the change in the average speed. The sliding process had a duration of approximately 30 s and could be divided into three stages: acceleration, deceleration, and stabilization. During the acceleration phase, the velocities of all particles increased sharply, and after approximately 10 s, the sliding velocity of the lower part of the accumulation layer reached a maximum of approximately 30 m/s. As frictional and collision energy

dissipated with the deformation of the accumulative layer, acceleration began to decrease. After 13 s, the average velocity of the accumulation layer reached a maximum of 26.87 m/s. At this time, the front edge of the accumulation layer entered the riverbed, and the sliding speed of the accumulation layer decreased markedly. As the particles slid along the sliding bed, the height of the accumulation layer in the valley gradually decreased. Finally, the sliding process ended after 33 s.

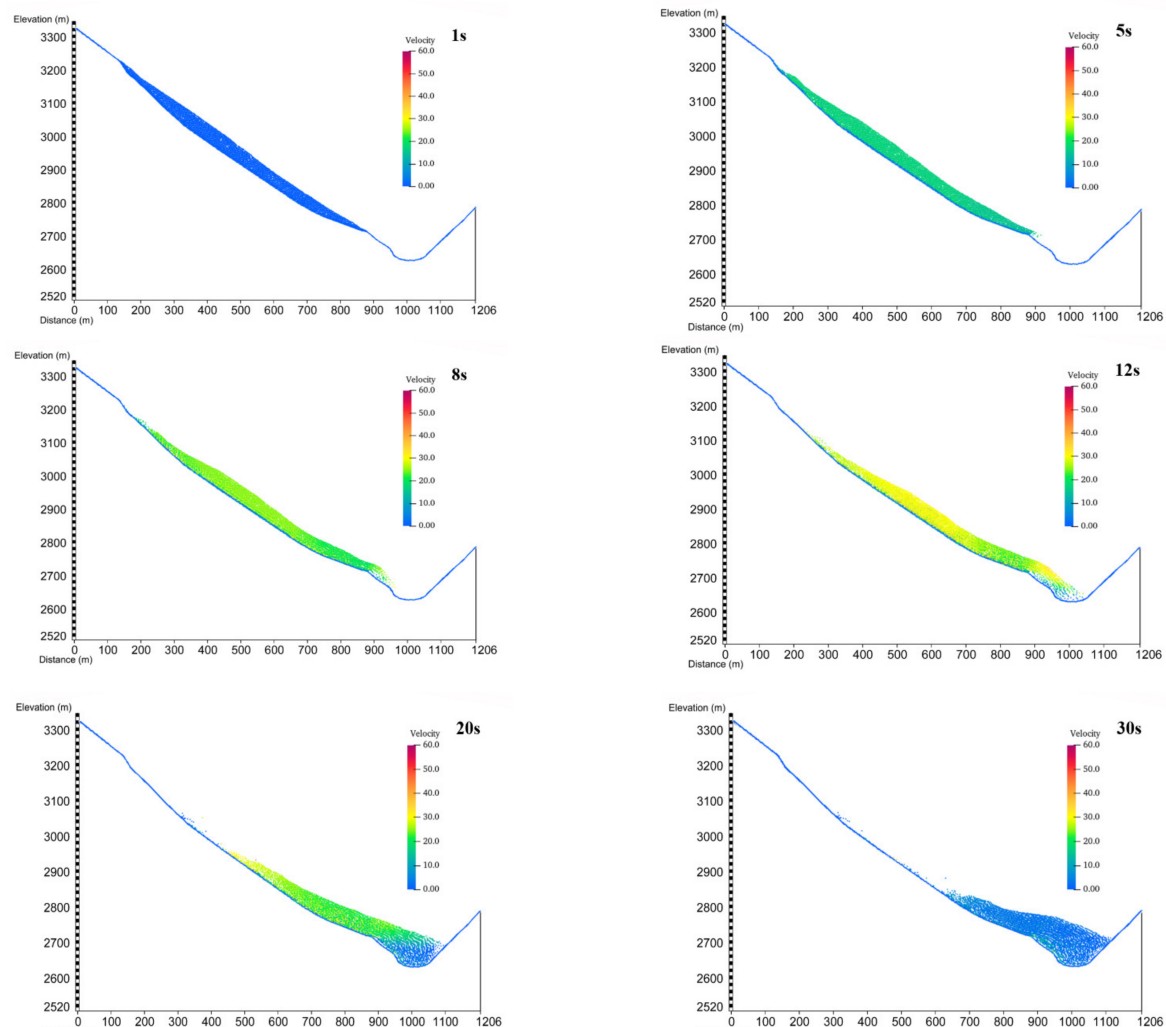

**Figure 11.** Cloud map of sliding velocity distribution in the accumulation layer at different times (1, 5, 8, 12, 20, and 30 s) during a landslide.

The changes in velocity displacement at the six monitoring points are shown in Figure 12a,b, respectively. Because of the sliding direction of the accumulation layer, in the early stage, the velocities measured at monitoring points 3 and 5 deeper within the accumulation layer were lower than those measured at monitoring points 4 and 6 in the shallower part of the accumulation layer. The velocity at deep monitoring point 1 on the trailing edge of the accumulation layer was greater than that at shallow monitoring point 2. Monitoring points 1, 4, and 6 recorded peak velocities of 31.63, 29.64, and 32.64 m/s after 14.5, 11.5, and 10.5 s, respectively. Monitoring point 3 recorded a peak velocity of 36.35 m/s after 17 s, considerably later than other monitoring points. The main reason for this was that monitoring point 3 was blocked by particles on the front edge during the sliding process. After the lower accumulation layer slid into the riverbed, sliding at monitoring point 3 accelerated. Based on these results, the Mohr–Coulomb model and MPM effectively

simulated the sliding process of the accumulation layer and intuitively represented the entire process of instability sliding, high-speed sliding, and deceleration.

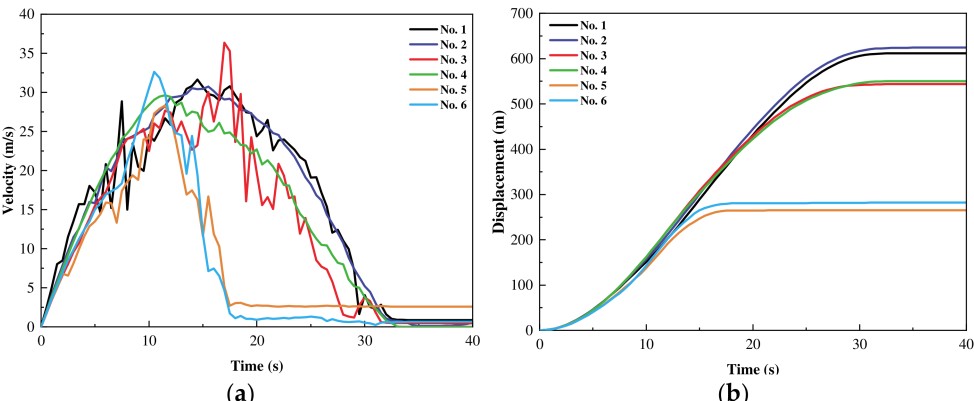

**Figure 12.** Changes in (**a**) velocity and (**b**) displacement of the accumulation layer at different monitoring points.

The displacement characteristics of an accumulation layer are closely related to landslide movement and accumulation [23]. In Reference [23], the landslide displacement simulated by the author rose rapidly in the first 30 s, and then gradually slowed down. In this study, the displacement of the body increased rapidly in the first 25 s and then increased slowly. As shown in Figure 13b, the maximum average displacement of the accumulation layer after 33 s of sliding was 477.71 m. The displacement of the accumulation layer increased rapidly between 0 and 13 s, then slowly decelerated. Among the six monitoring points, 5 and 6 required the shortest time to reach equilibrium, and the maximum displacement of 624.13 m was measured at monitoring point 2.

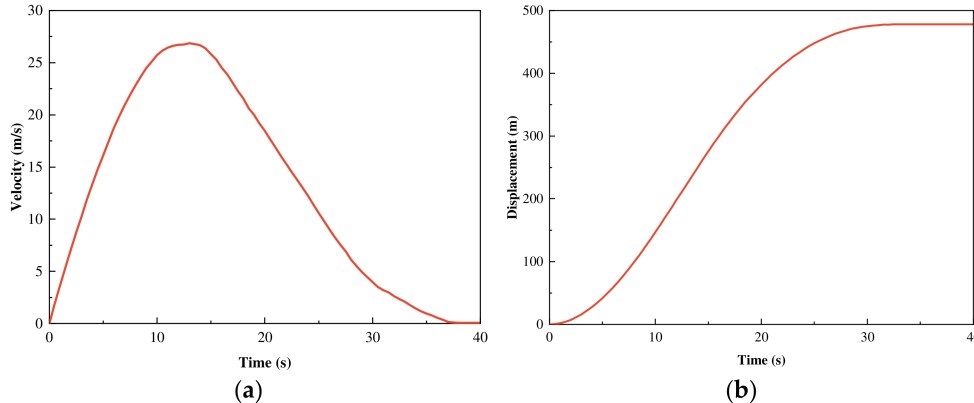

**Figure 13.** Average (**a**) sliding velocity and (**b**) sliding displacement of the accumulation layer.

As shown in Figure 14, after the landslide, the accumulation layer's maximum elevation was 2805 m, its maximum heights in the valley and on the opposite bank were 112 and 68 m, respectively, and its inclination was 15°. The dead water level of the reservoir was 2815 m. Impulse waves caused by the instability of the accumulation body and the risk of lake bursts formed by the accumulation body pose a major threat to dam construction, human safety, and property, and thus require special attention.

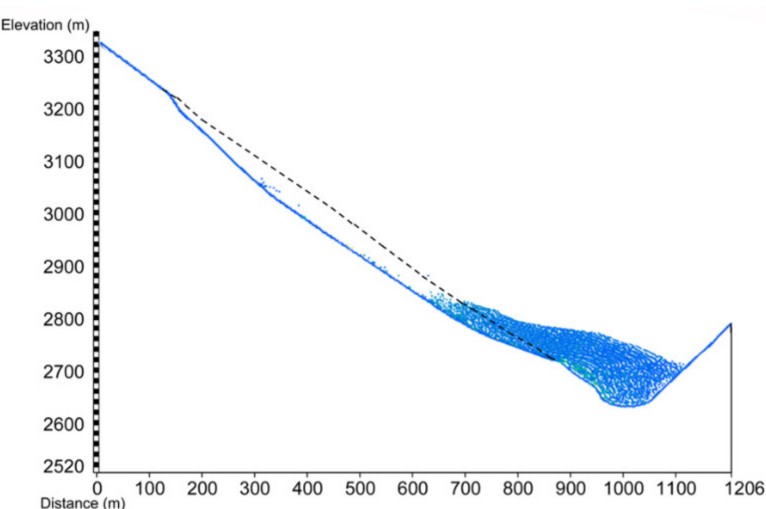

**Figure 14.** Post-landslide morphology of the accumulation layer.

## 4. Conclusions

In this study, the evolution and mechanisms of accumulation layer deformation, instability, and movement were analyzed in relation to changing reservoir water levels using a physical model test and numerical simulation. The main conclusions drawn from these methods are as follows:

(1) The pore water pressure in the accumulation layer increased and decreased with increasing and decreasing reservoir water level, respectively. The rate of change in pore water pressure at each measuring point was lower than that of the water level, indicating a degree of hysteresis.

(2) When the reservoir water level increased, the displacement of the accumulation layer increased slowly. When the reservoir water level dropped sharply, the displacement of the accumulation layer increased rapidly. The weak interlayer in the accumulation body cracked first, and then the cracks expanded and penetrated. Thus, the fluctuations in the reservoir water level damaged the front edge of the accumulation layer, leading to deformation and failure via the mode of traction-sliding.

(3) As the reservoir water level rose, the infiltration of the reservoir water into the accumulated soil caused the soil to soften and the shear strength to decrease. However, at this time, the change in the pore water pressure in the slope lagged behind the change in the reservoir water level outside the slope, resulting in the hydrodynamic pressure outside the slope pointing to the slope. The hydrodynamic pressure acting on the accumulation layer increased its stability. In contrast, as the reservoir water level dropped, the loss of hydrodynamic pressure acting on the accumulation layer decreased its stability. The faster the reservoir water level decreased, the greater the water pressure difference between the inside and outside of the accumulation layer. This resulted in a greater hydrodynamic pressure within the accumulation layer, pushing groundwater to the outer edge of the slope, decreasing its stability, and increasing the risk of a potential landslide.

(4) The accumulation layer sliding process could be divided into three stages: acceleration, deceleration, and stabilization. After the landslide, the accumulation layer's maximum elevation was 2805 m, its maximum heights in the valley and on the opposite bank were 112 and 68 m, respectively, and its inclination was 15°. The risk of dam bursting due to the lake formed by the accumulation of material during landslides poses a major threat to dam construction, human safety, and property, which requires special attention.

Based on the above findings, this study demonstrated how the combination of a physical and MPM-based numerical model can be used to effectively simulate the entire sliding process of an accumulation layer, from instability to high-speed sliding and deceleration. Therefore, this study serves as a reference point for future landslide research and associated risk assessment and prevention measures.



**Author Contributions:** Data curation, J.W.; writing—original draft, Y.W.; visualization, R.W.; project administration, R.C. and Z.W. All authors have read and agreed to the published version of the manuscript.

**Funding:** This work was funded by the National Natural Science Foundation of China (Grant No. 51939004), Fundamental Research Funds for the Central Universities (Grant No. B210203003) and Special Fund for Science and Technology Project of Guizhou Provincial Water Resources Department (Grant No. KT202229).

**Data Availability Statement:** The data that support the findings of this study are available from the corresponding author, [Wang R], upon reasonable request.

**Acknowledgments:** We are deeply indebted and owe our thanks to PowerChina Guiyang Engineering Corporation Ltd. for their assistance in the collection of engineering and geological survey data.

**Conflicts of Interest:** The authors declare that they have no conflict of interest.

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
