# Peer review of "Physical and Numerical Simulation of the Mechanism Underpinning Accumulation Layer Deformation, Instability, and Movement Caused by Changing Reservoir Water Levels"

_water, doi:10.3390/w15071289_

Round 1

Reviewer 1 Report

Line 13: revise “To reveal the mechanism underpinning the deformation”.

Line 36: “fourth-series” means “Quaternary”?

Line 36: revise “Hydrodynamic pressure landslides”.

Line 50: revise “simulation the slope stability”.

In the introduction, only the advantages of physical models and MPM method are shown. The shortcomings of existing researches need to be pointed out from the aspects of physical models and numerical simulations, especially for the pinning accumulation layer.

Line 95: The model test did not consider the pining in the accumulation layer.

Line 117: revise “Ct − Ck − Cv −;”.

Table 1: It seems that the similarity ratio of cohesion did not agree with the ratio in the text "200" Line 117. Please explain more.

Line 239: How did the author consider the actions of pore water pressure and reservoir water level in the MPM simulation? They greatly affect the movement process.

Line 240: If the MPM method is used to simulate the movement of landslide, the text should explicitly describe the purpose of the numerical simulation.

Line 298: Mohr-Coulomb?

Line 302-303: Please make comparison between the simulated data and literature [18].

Reviewer 2 Report

As reflected in the title, this paper focuses on accumulation layer deformation and movement caused by the water level and conducted the basic physical model and MPM simulation. This paper is good for the readers to understand the effect of water level on the slide of accumulation layer. To my opinion, the paper is less successful in innovation, but this experimental data is worthy to be payed attention to. My comments are presented in the following.

1.     Lines 52-58 in the introduction. the author didn’t make it clear what your innovation. Did other research conduct the similar physical model or simulation? If they did, please emphasize the difference between your research and others.

2.     Lines 97-101, the authors should make detailed information about the physical model. How to set the boundary geometry between bedrock and sliding body? Did you consider the geometry effect on the whole research?

3.     For caption in the figure 3, the physical meaning of symbols ‘P1’,’P2’,’P3’ and ‘D1’, ‘D2’, ‘D3’ and ‘D4’ be written in the caption, which will be easy for the readers to understand the figure.

4.      This paper is integrated research. The author needs to emphasize the new finding or tell the readers what they can study from this paper. The physical model and simulation had been studied in many published papers. Suggest that the authors revised the abstract and conclusion to make it clear to readers that what the new finding and highlight is.

Reviewer 3 Report

The paper utilizes experimental and numerical simulation methods to investigate the hydrodynamic pressure landslides. The authors present many interesting findings. Overall this research is well-designed and presented. I would recommend publishing the paper after a minor revision. Details comments were listed below:

1.       In the introduction, the authors did not review enough previous studies to clarify the novelty which differed from the previous study.

2.       In table 2, why did the same parameters repeat twice?  Can the authors include more details on the test procedures?

3.       “Based on these changes in water level, pore water pressure and displacement were measured at each of their respective measuring points to obtain associated change curves.” The authors should include the methods on how to measure the pore water pressure and displacement.

4.       In Figure 5, can the authors mark the point of each pressure change to better understand the changes corresponding to which stage of the test procedure?

5.       If the part of the Numerical Model Description (3.2.1) can be reorganized, and merged into the part of the Numerical Model of the Accumulation Layer Landslide (2.3), it would be better for readability.

6.       “The sliding process had a duration of approximately 30 s and could be divided into three stages: acceleration, deceleration, and stabilization.” Please explain the classification criteria. Did previous studies propose that similar perspective? The author should compare the findings with the prior study in the part of the results of discussions.

7.       “Figure 12, Changes in (a) velocity and (b) displacement of the accumulation layer at different monitoring points.” Please explain how the authors measure the velocity at different monitor points. 

Round 2

Reviewer 2 Report

I think it could be accepted for publication.